# Scaling Laws in Patchification: An Image Is Worth 50,176 Tokens And More

Feng Wang [1]   Yaodong Yu [2]   Wei Shao [3]   Yuyin Zhou [4]   Alan Yuille [1]   Cihang Xie [4]

## Abstract

Since the introduction of Vision Transformer (ViT), patchification has long been regarded as a *de facto* image tokenization approach for plain visual architectures. By compressing the spatial size of images, this approach can effectively shorten the token sequence and reduce the computational cost of ViT-like plain architectures. In this work, we aim to thoroughly examine the information loss caused by this patchification-based compressive encoding paradigm and how it affects visual understanding. We conduct extensive patch size scaling experiments and excitedly observe an intriguing scaling law in patchification: the models can consistently benefit from decreased patch sizes and attain improved predictive performance, until it reaches the minimum patch size of $1 \times 1$, *i.e.*, pixel tokenization. This conclusion is broadly applicable across different vision tasks, various input scales, and diverse architectures such as ViT and the recent Mamba models. Moreover, as a by-product, we discover that with smaller patches, task-specific decoder heads become less critical for dense prediction. In the experiments, we successfully scale up the visual sequence to an exceptional length of 50,176 tokens, achieving a competitive test accuracy of 84.6% with a base-sized model on the ImageNet-1k benchmark. We hope this study can provide insights and theoretical foundations for future works of building non-compressive vision models. Code is available at https://github.com/wangf3014/Patch_Scaling.

## 1. Introduction

In the past few years, we have witnessed the great success of Vision Transformers (ViTs) in representation learning, with a series of visual foundation models learned with this plain architecture achieving highly competitive performance and establishing effective connections to other modalities such as natural language (Dosovitskiy et al., 2021; Caron et al., 2021; Radford et al., 2021; Yu et al., 2022a; Rombach et al., 2022; Kirillov et al., 2023; Liu et al., 2023). A key insight behind the ViT-like architectures lies in a compressive encoding paradigm: instead of directly processing raw pixels that introduces significant complexity, these architectures leverage a patchification layer to compress images into spatially smaller feature maps, making the representation space of an image roughly equivalent to that of a medium-length text consisting of a few hundred tokens.

However, we argue that this operation often incurs irreversible information loss to visual inputs. For example, intuitively, we believe the information contained in a $224 \times 224$ resolution image is generally much richer than that in a text consisting of 196 words; however they have nearly the same size of representation space under a ViT encoder with patch size $16 \times 16$ (we suppose the vision and language encoders share the same embedding dimension). The difference in information content between visual and textual data can also be directly reflected in their storage requirements: storing an uncompressed 24-bit, $224 \times 224$ resolution image requires approximately 147KB, whereas storing a 196-word text only needs about 1.15KB. Empirically, if we manually reduce the compression rate, for example, by changing the patch size of DeiT-Base from $16 \times 16$ to $8 \times 8$, we can observe a significant accuracy improvement from 81.8% to 83.5% on the ImageNet-1k classification benchmark (Deng et al., 2009).

Nonetheless, since the computation of self-attention scales quadratically with sequence length, ViT architectures are sensitive to the patch size. At the time when ViT was first introduced in late 2020, it needed to ensure that its computational cost was comparable to that of the CNN counterparts; and given the computational capacity at that time, the models had to be computationally manageable in terms of memory consumption and training time, especially when trained with the medium-resolution, medium-scale ImageNet (Deng et al., 2009) and beyond (Sun et al., 2017). As a result, the architectural design of ViT had to compromise with a compressive encoding paradigm achieved through patchification. The success of this design, patchification with a typical $16 \times 16$-pixel kernel, has led to its widespread adoption as

---

[1]Johns Hopkins University [2]UC Berkeley [3]University of Florida [4]UC Santa Cruz. Correspondence to: Cihang Xie <cixie@ucsc.edu>.

*Proceedings of the 42$^{nd}$ International Conference on Machine Learning*, Vancouver, Canada. PMLR 267, 2025. Copyright 2025 by the author(s).

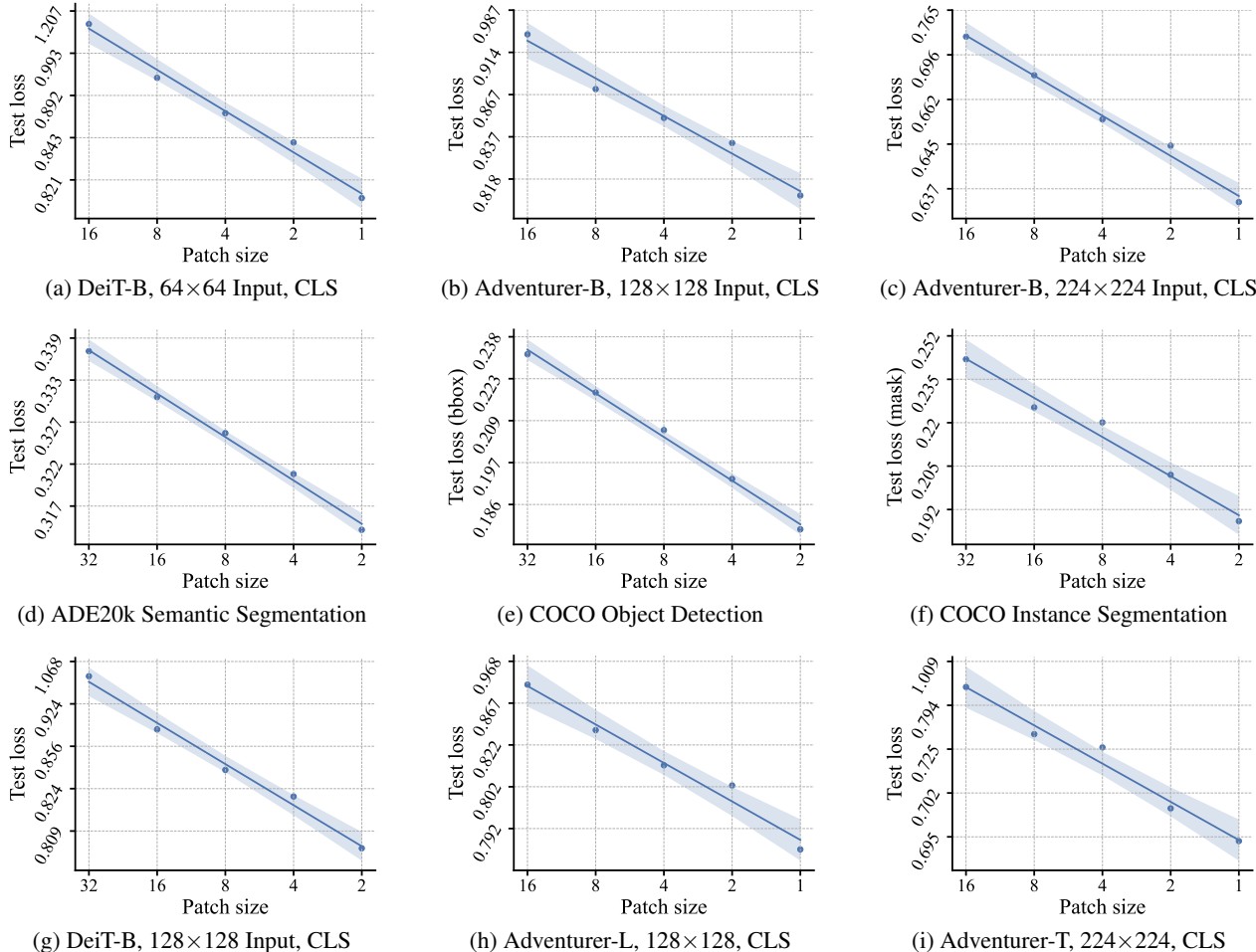

Figure 1. **Patchification Scaling Laws.** We observe a smooth and consistent decrease in test loss across different vision tasks, input resolutions, and model architectures when reducing the patch size. The performance gains remain considerably significant even when scaling down the patch size to 1×1. In all sub-figures, both $x$ and $y$ axes are in log scale. CLS denotes ImageNet-1k classification.

a default component in various subsequent architectures, even including those non-attention models such as ConvNeXt (Liu et al., 2022) and Vision Mamba (Zhu et al., 2024), while the impact of information loss posed by this compressive encoding paradigm has not been well studied.

In this work, we aim to thoroughly examine how compressive encoding affects visual representations and whether patch size can be a new scaling dimension for modern visual architectures. While the concept of Scaling Laws (Kaplan et al., 2020) has been broadly testified in natural language processing, leading to a great prosperity of Large Language Models over the past few years (Touvron et al., 2023; Team et al., 2023; Achiam et al., 2023), the scaling-up of vision models faces practical issues in the dimensions of both parameter size and input size (detailed in Section 4.3). Here, we aim to revisit the scaling potential of vision models from a new perspective of spatial compression, attempting to un-

lock the compressed information by reducing the patch size. We highlight that through patchification, there is significant room for scaling up the model's computation, and a new scaling law may emerge during this process.

Thanks to the rapid advancements in hardware, efficient attention mechanisms (Dao et al., 2022; Kwon et al., 2023), as well as linear-complexity structures (Katharopoulos et al., 2020; Peng et al., 2023; Gu & Dao, 2023), we can now extensively validate the impact of patchification at a standard input size (*e.g.*, 224×224 for ImageNet) with manageable computing resources. We conduct a series of straightforward scaling experiments on patchification, gradually reducing the model's patch size from the typical 16×16 down to 1×1 to lower the compression rate and observe how performance changes. We employ both ViT and Adventurer (Wang et al., 2024c), a Mamba-based (Dao & Gu, 2024) linear-complexity architecture, to make our conclu-

sions generalizable and experiments affordable in computation. To our surprise, this simple scaling study delivers three intriguing discoveries:

First, as shown in Figure 1, we excitedly observe a new scaling law for patchification in vision models. Similar to the scaling laws discovered in early studies (Kaplan et al., 2020) on language—where continuously increasing model parameters reliably leads to consistent performance gains—we have identified a new scaling dimension for vision models, which is reflected in the observation that as the compression rate (*i.e.*, patch size) decreases, the model's test loss smoothly declines, reaching its limit at single-pixel patch sizes that essentially form a non-compressive encoding paradigm. This conclusion broadly holds true for various vision tasks, diverse input scales, and different visual architectures.

Second, we confirm that visual encoding can be performed in a very long token sequence, while patchification is not a requisite for building effective vision models, but rather a compromise to memory and computation overhead when resource is limited. The information lost in the compression of the patchification layer is actually crucial for the model's prediction: on the standard $224 \times 224$-resolution ImageNet-1k classification benchmark, we remove the patchification operation and form a super-long visual sequence consisting of 50,176 tokens, by which we boost the model's test accuracy from 82.6% to a remarkable result of 84.6%.

Finally, we observe a compelling phenomenon in semantic segmentation: as we transit from patch-based tokenization to pixel-level modeling, the traditional necessity for a decoder head—long considered a default component since the inception of deep network architectures—can be eliminated without compromising performance. This architectural simplification is potentially profound, suggesting the possibility of developing decoder-free dense prediction models and illuminating the path toward a universal, encoder-only visual architecture capable of learning from every pixel.

## 2. Related Work

**Generic visual backbones.** The development of visual backbones has fundamentally shaped the field of computer vision. Initially dominated by Convolutional Neural Networks (CNNs), these architectures have evolved to gain increasing capabilities for visual representation learning. Pioneering works such as LeNet (LeCun et al., 1998) and AlexNet (Krizhevsky et al., 2012) have proven the significant effectiveness of convolutional architectures in large-scale image classification tasks. Following these foundational models, the architecture has been refined with the innovations in model depth (Simonyan & Zisserman, 2015), residual connection (He et al., 2016; Huang et al., 2017), and efficient neural architecture search (Tan & Le, 2019).

The landscape of visual backbones underwent another round of significant transformation with the introduction of ViTs (Dosovitskiy et al., 2021) in late 2020, where a novel plain architecture was proposed that treats images akin to language sequences. This model utilizes a simple patchification layer to convert images into sequences of tokens, which are then processed using mechanisms adapted from language models. This approach opened new avenues in handling visual data without the inductive biases inherent in CNNs, demonstrating competitive performance on several benchmarks. The success of ViTs have spurred rapid development and innovations in data-efficient training strategies (Touvron et al., 2021a; 2022a;b), self-supervised learning techniques (Caron et al., 2021; Chen et al., 2021b; Bao et al., 2022; He et al., 2022; Ren et al., 2024b), vision-language understanding (Radford et al., 2021; Jia et al., 2021; Liu et al., 2023; Alayrac et al., 2022; Yu et al., 2022a; Wang et al., 2024a), and hierarchical architecture designs (Liu et al., 2021; Chen et al., 2021a; Yuan et al., 2021; Yu et al., 2022b).

Inspired by the patchification design of transformers, there have been many CNN-based (Liu et al., 2022) and State Space Model (Kalman, 1960; Gu et al., 2022; 2021) based architectures (Zhu et al., 2024; Wang et al., 2024b;c) following the same paradigm. Notably, the Mamba (Gu & Dao, 2023; Dao & Gu, 2024) token mixer, due to its advantage of linear complexity, has recently been widely used to explore vision tasks and has achieved competitive results (Zhu et al., 2024; Wang et al., 2024b;c; Ren et al., 2024a; Liu et al., 2024; Yang et al., 2024; Hatamizadeh & Kautz, 2024; Huang et al., 2024; Li et al., 2024; Ren et al., 2024c; Lieber et al., 2024; Wei & Chellappa, 2025). Among them, the Adventurer (Wang et al., 2024c) architecture, which significantly simplifies the overall model, has demonstrated superior speed compared to the Transformer. In this paper, we employ it as one of the primary experimental models.

**Visual architecture scaling.** Scaling laws was initially studied in natural language processing (Kaplan et al., 2020). In vision, a similar concept has guided the community to scale up foundational models in both parameter size and data volume. For example, in the age of CNNs, EfficientNets (Tan & Le, 2019; 2021) have proposed to scale-up the models in depth, width, and resolution. These advancements were then integrated into ResNets, leading to nearly Billion-level parameter CNNs (Xie et al., 2020; Kolesnikov et al., 2020; Huang et al., 2019; Bello et al., 2021; Wightman et al., 2021). Scaling the parameter count of Vision Transformers has also shown a great success in modern visual understanding benchmarks and has exhibited state-of-the-art results (Touvron et al., 2021b; Zhou et al., 2021; Zhai et al., 2022; Dehghani et al., 2023). More recently, Nguyen et al. introduce Pixel Transformers, with standard patchification grids scaled down to pixels, showcasing promising scaling

results for low-resolution (*e.g.*, 32×32) input images.

## 3. Method

### 3.1. Problem Formulation

This work aims to investigate the impact of spatial compression on the representation capability of modern visual architectures by scaling the downsampling rate of the patchification operation. The primary experiments are conducted on ViT-like plain architectures, with their definition as follows: The image encoder $\boldsymbol{F} : \mathbb{R}^{3 \times w \times h} \to \mathbb{R}^{L \times D}$ consists of a patchification layer at the beginning, positional embeddings, and a number of cascade token mixers and channel mixers. The patchification layer divides the input image $\boldsymbol{x} \in \mathbb{R}^{3 \times w \times h}$ into non-overlapping patches of size $p \times p$, flattening and projects them into a 1D token sequence $\boldsymbol{x}' \in \mathbb{R}^{L \times D}$. The following mixer layers extract deep visual features while keeping the sequence length $L$ and the feature dimensionality $D$ unchanged—which means the patchification layer makes the only spatial compression throughout the whole visual encoder.

To eliminate the influence of different mixer types on the results of patchification scaling, we conduct the main experiments using two visual encoders: the standard ViT (Dosovitskiy et al., 2021) and Adventurer (Wang et al., 2024c). Due to the significant memory and computation challenges posed by the quadratic complexity of self-attention, ViT is only used for context lengths within 4,096 in this work. For longer sequence tasks, we employ Adventurer, a recent Mamba-based (Gu & Dao, 2023; Dao & Gu, 2024) efficient architecture that excels in modeling long range dependencies with linear complexity. Adventurer shares the same plain framework as ViT, with spatial compression only presents in the initial patchification layer, while the key difference is that Adventurer leverages the recent Mamba (Dao & Gu, 2024) module as its token mixer, which has a linear complexity relative to sequence length and allows us to perform pixel tokenization for even the standard 224×224 resolution inputs within reasonable computational resources (*e.g.*, 256 A100 GPUs). Remarkably, in our experiments, we form a super-long visual sequence of 50,176 tokens for ImageNet inputs by scaling down the patch size to 1×1.

### 3.2. Technical Details

We conduct patchification scaling experiments on image classification, semantic segmentation, object detection and instance segmentation tasks. Following the standard design of ViTs (Dosovitskiy et al., 2021) and Adventurer, we extract holistic visual features by a learnable [CLS] token for classification. For object detection and instance segmentation, we load backbones pretrained with classification and employ a Cascade Mask R-CNN (Cai & Vasconcelos, 2019)

as decoder head. Note that we use the same patch size for classification pretraining and downstream finetuning to ensure consistency in the scaling property.

For semantic segmentation, in addition to evaluating the standard encoder-decoder structure, we also explore a decoder-free approach to observe the emerging properties of patchification scaling. Specifically, instead of using a deep UperNet (Xiao et al., 2018) as the default segmentation head, we employ a simple linear layer to project the dense features extracted by the backbone into the category dimension for training the semantic segmentation task.

This modification is based on the following prior assumption: in dense prediction tasks like semantic segmentation, the decoder head serves two main functions. The first is addressing the issue where the backbone's high downsampling rate results in feature granularity that is insufficient for pixel-level predictions—typically mitigated by designs such as atrous convolution and multi-scale feature fusion (Chen et al., 2017; 2018). The second function is enhancing the model's learning capacity by introducing additional trainable parameters. Under this assumption, we believe that if the backbone's compression rate is already very low, the decoder's benefits would be limited to the second aspect. Therefore, task-specific decoder head designs become less critical, and training a general high-fidelity backbone alone would be sufficient to handle various vision tasks.

## 4. Experiments

### 4.1. Experimental Setup

The experiments are conducted on the standard ImageNet-1k (Deng et al., 2009) classification, ADE20k (Zhou et al., 2019) semantic segmentation, and COCO (Caesar et al., 2018) object detection and instance segmentation benchmarks. For ViTs, we follow the data-efficient strategy of DeiT (Touvron et al., 2021a) to train the model for 300 epochs by an AdamW (Loshchilov & Hutter, 2019) optimizer with a 1024 batch size, 0.001 learning rate and 0.05 weight decay. For Adventurer, we basically refer to their optimized multi-stage training recipe to improve efficiency and obtain competitive results. The details of the training strategy can be found in Appendix. In semantic segmentation, we follow the prior practice of DeiT and Adventurer to finetune the classification models with an AdamW optimizer, 5e-5 learning rate, 0.01 weight decay, a total batch size of 16 for 160k iterations. We train object detection and instance segmentation with AdamW optimizer, 1e-4 learning rate and 0.05 weight decay for 12 epochs.

### 4.2. Main Results

As shown in Figure 1, we first evaluate the model's patchification scaling performance using test loss as a unified metric

| Model | Input size | patch tokenization | | | | pixel tokenization | |
|---|---|---|---|---|---|---|---|
| | | p=16 | p=8 | p=4 | p=2 | p=1 | seq. length |
| DeiT-Base (Touvron et al., 2021a) | 64×64 | 68.2 | 76.9 | 80.1 | 80.8 | **81.3** | 4,096 |
| DeiT-Base (Touvron et al., 2021a) | 128×128 | 78.1 | 81.0 | 82.3 | 82.9 | - | - |
| Adventurer-Base (Wang et al., 2024c) | 64×64 | 69.2 | 77.2 | 80.0 | 80.5 | **80.9** | 4,096 |
| Adventurer-Base (Wang et al., 2024c) | 128×128 | 79.0 | 81.5 | 81.8 | 82.2 | **82.4** | 16,384 |
| Adventurer-Base (Wang et al., 2024c) | 224×224 | 82.6 | 83.9 | 84.3 | 84.5 | **84.6** | **50,176** |

*Table 1.* **Detailed ImageNet classification results.** As patch size (denoted as *p*) decreases, the test accuracy (%) on ImageNet-1k (Deng et al., 2009) consistently improves and reaches the best performance with pixel tokenization. We highlight that we successfully scale up the visual token sequence to an unprecedented length of 50,176, with a competitive 84.6 test accuracy obtained by a base-sized model.

across different input sizes, tasks, and parameter scales. We observe an interesting phenomenon that the model's predictive performance consistently improves as the patch size decreases. This observation effectively highlights the negative impact of the existing compressive encoding approach in visual models and supports our initial hypothesis: patchification is not a necessary component for visual encoders; its primary role is to improve computational efficiency at the cost of partial information loss. Although this efficiency gain is significant for Transformer models with quadratic complexity, our findings suggest that when the computing resource allows—and indeed, computational power has evolved rapidly over years—we should reconsider the traditional compressive encoding approach and begin embracing the notion of *"a pixel is worth a token"* that stands for a non-compressive representation learning paradigm.

We also observe that reducing the patch size not only improves performance in dense prediction tasks like semantic segmentation and instance segmentation—which naturally favor fine feature granularities and for which smaller patch size is a direct solution—but also benefits holistic tasks like image classification, which inherently do not require fine-grained representations. This result indicates that the primary benefit of reducing the patch size comes from unlocking the visual information that is previously compressed by patchification. This information, often considered insignificant low-level features in the past, is actually considerably critical for visual understanding.

**ImageNet classification** results are elaborated in Table 1. As shown, in terms of test accuracy, the models also experience a smooth and consistent performance improvement with patch size decreasing. Notably, with the help of Adventurer's linear time complexity and efficient memory consumption, we successfully scale up the visual token sequence to a length of 50,176 in the ImageNet classification task. To our knowledge, this is the *first time* that modern visual architectures have extended the input sequence to such a length and processed it directly without partitioning. It not only achieves a highly competitive 84.6% test accuracy with a base-sized model (100M parameters), but more importantly, it demonstrates that visual understanding can

| Model | Decoder | Params | Patch size | mIoU |
|---|---|---|---|---|
| | *UperNet* | *17M* | *16×16* | *41.3* |
| | None | 12M | 16×16 | 40.0 |
| Adventurer-T | None | 12M | 8×8 | 41.6 |
| | None | 13M | 4×4 | 42.1 |
| | None | 13M | 2×2 | **42.5** |
| | *UperNet* | *112M* | *16×16* | *45.7* |
| | None | 99M | 16×16 | 44.0 |
| Adventurer-B | None | 99M | 8×8 | 45.5 |
| | None | 100M | 4×4 | 46.3 |
| | None | 100M | 2×2 | **46.8** |

*Table 2.* **ADE20k semantic segmentation.** We focus on decoder-free structures and observe the mIoU score improves smoothly when patch size shrinks. We highlight the results that reach the limits of hardware capabilities in blue and best results **bolded**.

be effectively performed from very long contexts.

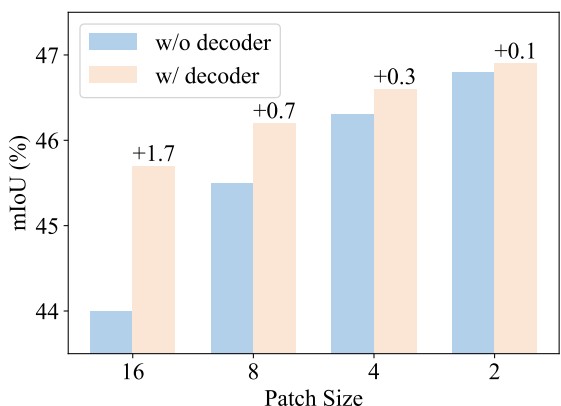

*Figure 2.* **Decoder's impact on semantic segmentation.** We train a semantic segmentation model with the same backbone but different decoder heads: an UperNet with 13M parameters and a simple linear layer with 0.2M parameters. We observe that as patch size decreases, the impact of the decoder head diminishes.

**ADE20k semantic segmentation** results are summarized in Table 2. As shown, we observe the same scaling behavior in this dense prediction task, with its test loss smoothly decreasing (see Figure 1) and mIoU score consistently improving as patch size shrinks. It is worth noting that even though we eliminate the task-specific decoder head in this

| Model | Patch | $AP^b$ | $AP^b_{50}$ | $AP^b_{75}$ | $AP^m$ | $AP^m_{50}$ | $AP^m_{75}$ |
|---|---|---|---|---|---|---|---|
| Adventurer-T | 32×32 | 44.7 | 63.3 | 48.6 | 38.4 | 60.4 | 41.4 |
| | 16×16 | 46.5 | 65.2 | 50.4 | 40.3 | 62.2 | 43.5 |
| | 8×8 | 48.0 | 66.7 | 51.8 | 41.7 | 63.6 | 45.0 |
| | 4×4 | 48.5 | 67.1 | 52.3 | 42.2 | 64.1 | 45.4 |
| | 2×2 | **48.7** | **67.3** | **52.4** | **42.4** | **64.3** | **45.7** |
| Adventurer-B | 64×64 | 44.1 | 62.8 | 48.0 | 38.3 | 60.1 | 41.8 |
| | 32×32 | 46.4 | 65.0 | 50.3 | 40.6 | 62.5 | 43.1 |
| | 16×16 | 48.4 | 67.2 | 52.4 | 42.0 | 64.8 | 45.0 |
| | 8×8 | 49.5 | 67.9 | 53.3 | 42.9 | 65.5 | 46.1 |
| | 4×4 | **50.3** | **68.5** | **54.0** | **43.4** | **66.0** | **46.6** |

*Table 3.* **COCO object detection and instance segmentation.** Similar to classification and semantic segmentation results, these two tasks exhibit consistently enhanced performance as patch size decreases. We highlight the results that reach the limits of hardware capabilities in blue and best results **bolded**.

experiment, the encoder-only models—whether the 13M-parameter tiny-sized model or the 100M-parameter base-sized model—can still produce competitive results when the encoding compression rate becomes sufficiently low.

Figure 2 presents a direct comparison on the impact of decoders in semantic segmentation, where we load the same pretrained backbone (Adventurer-Base) and finetune it separately with a UperNet (Xiao et al., 2018) and a simple linear projection layer. As shown, with a high spatial compression rate such as 16×, the model can easily benefit from a decoder head; however, as the patch size decreases and the encoder itself can produce sufficiently fine-grained features, the functionality of decoders starts to be marginalized.

Interestingly, this experiment validates our hypothesis presented in Section 3.2, demonstrating that the core component of developing dense prediction models lies in reducing the spatial compression rate, while the help that decoder heads can provide is very limited. This insight further suggests that with non-compressive encoders, it becomes feasible to build a visual foundation model that could provide pixel-level representations and effectively supports various downstream tasks without requiring significant efforts to adapt to their specific objectives. In this work, we keep focusing on the exploration of patchification scaling and leave the development of ***pixel foundation models*** for future research. We hope that our findings here can provide a solid theoretical foundation for such endeavors.

**COCO object detection and instance segmentation** tasks also showcase a similar effect of patchification scaling. As summarized in Table 3, both tasks achieve their best performance when the patch size reaches the hardware's computational limits (2×2). Compared to the high compression baselines, both Adventurer-Tiny and Base models demonstrate significant precision improvements, such as 48.7% *vs.* 44.7% for Adventurer-Tiny and 50.3% *vs.* 44.1% for Base.

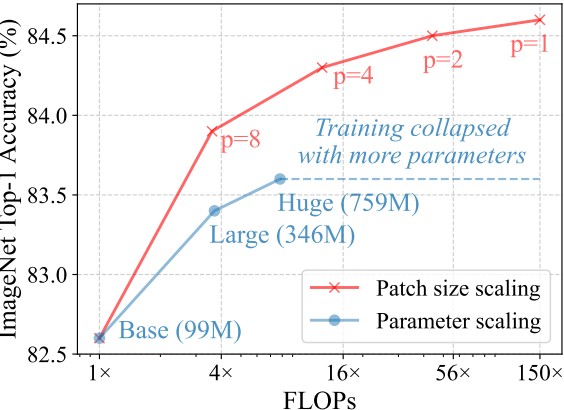

(a) Scaling from Adventurer-Base/16, 224×224 input.

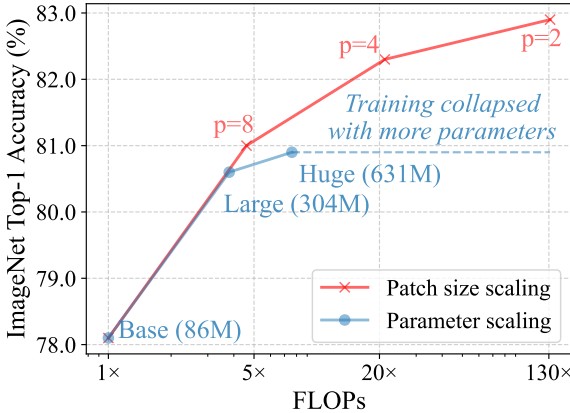

(b) Scaling form ViT-Base/16, 128×128 input.

*Figure 3.* **Patch size scaling *vs.* parameter scaling.** Given an Adventurer-Base with 224×224-resolution inputs, we scale up the model along two dimensions respectively. The model struggles to achieve further accuracy improvements beyond ∼760M parameters, whereas scaling down the patch size continues to show a consistent upward trend in performance.

Notably, we have conducted patch size scaling experiments across four tasks: object classification, semantic segmentation, object detection, and instance segmentation. These experiments span a variety of input resolutions (from 64×64 to a short side of 800), different training objectives, and different token mixer types (self-attention and Mamba (Gu & Dao, 2023)). Despite these variations, a consistent and generalizable conclusion emerges: ***Reducing patch size reliably guarantees performance gains***.

### 4.3. Ablation Studies

**Patchification scaling *vs.* parameter scaling.** In Figure 3a, we compare the impact of scaling down the patch size versus scaling up the parameter count on the performance of Adventurer. As shown, for a fixed patch size and input

size, increasing the parameter count within a certain range (*e.g.*, up to 760M parameters) yields significant performance gains. However, further scaling beyond this point does not necessarily lead to additional benefits. In fact, overcoming the parameter scaling bottleneck in vision models is both technically challenging and costly. It often requires investing in higher-quality training data (Zhai et al., 2022; Radford et al., 2021), incorporating self-supervised learning approaches (Caron et al., 2021; He et al., 2022), and making extensive hyperparameter tuning efforts (Touvron et al., 2022b).

In contrast, patch size scaling not only exhibits a better computation-accuracy tradeoff and achieves higher performance limits than parameter scaling, but it also offers a simpler and more straightforward learning process: when training with different patch sizes, there is no need to modify training strategies or datasets, and all experiments can be done in a single run using the same set of hyperparameters.

The potential of patchification scaling is even more evident in ViT. With the same input scale, reducing the patch size yields greater performance improvements for ViT compared to the linear-complexity Adventurer. Additionally, in terms of FLOPs, ViT has more room for scaling, as its computation grows quadratically with sequence length. As shown in Figure 3b, due to this quadratic complexity, ViT experiences a larger increase in FLOPs than Adventurer when scaling down the patch size, leading to a similar accuracy growth over FLOPs as that of parameter scaling (*e.g.*, ViT-Base/8 *vs.* ViT-Large/16). However, when investing higher FLOPs, parameter scaling falls significant short and may easily collapse with higher parameter counts.

**Limitations of input size scaling.** Compared to scaling down the patch size at a fixed input size, another method—directly scaling up the input size—can achieve a similar effect of reducing the compression rate and extending the token sequence. However, we contend that changing the input size is not a flexible and applicable approach for effective scaling, as its upper bound is easily constrained by the original resolution of the image. For example, in the standard ImageNet classification benchmark, images are resized to 224×224 for both training and evaluation stages (Touvron et al., 2021a; Liu et al., 2022; Wang et al., 2024c). This input size has actually compressed the visual information, as the average ImageNet image size is approximately 490×430 pixels. Within this range, scaling the input size is generally more effective than scaling the patch size. For example, Adventurer-B/16 with a 448×448 input outperforms Adventurer-B/8 with a 224×224 input by 0.4%, despite having similar parameter counts and FLOPs.

However, beyond this input size, further increasing the input dimensions provides diminishing returns in performance gains. If we fix the sequence length—scaling up the in-

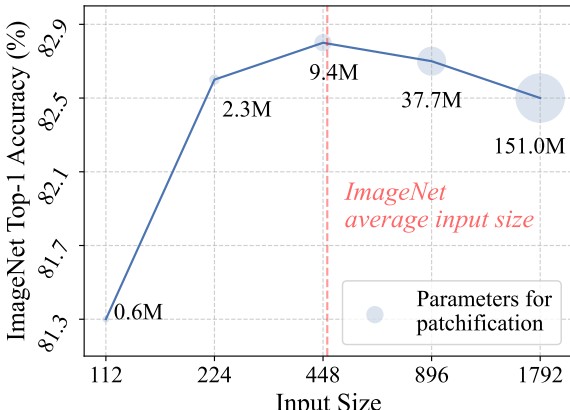

*Figure 4.* **Input size scaling with fixed sequence length.** We fix the ratio of $image\_size/patch\_size$ and scale up the input size for ImageNet classification. As shown, when the input size is scaled beyond its original resolutions (*e.g.*, typically 460 for ImageNet), further interpolating the input images does not yield additional accuracy gains. Instead, it leads to a rapid increase in patchification parameters, resulting in training instability that ultimately harms performance.

put size while proportionally increasing the patch size—the model undergoes a rapid growth in parameter count in the patchification layer, which may easily result in reduced model efficiency and stability during scaling. We showcase this issue in Figure 4, where it is observed that resizing inputs beyond their original resolutions does not provide additional information gains. Instead, the over-parameterization of the patchification layer leads to a considerable performance degradation. As comparison, the direct scaling of patch size can effectively avoid the over-parameterization issue, making it a flexible and practical scaling dimension for modern visual architectures.

**Scaling in both dimensions.** In Table 4, we provide more ImageNet classification results with Adventurer models, where we scale them in both model size (parameter count) and patch size. As shown, the two scaling dimensions work synergistically when offering performance gains, with the highest accuracies consistently achieved by either the largest models or the smallest patch sizes. As analyzed earlier, the function of patch size scaling lies in reducing the spatial compression rate and thereby enabling the extraction of richer information from the data itself. Intuitively, this effect does not conflict with scaling up the model size, where performance gains mainly stem from enhanced fitting capabilities provided by increased parameters.

The results in this experiment suggest that, given sufficient computing resources, we can easily transfer past advancements in parameter scaling to patchification scaling. In other words, patchification scaling can serve as a complement to model size scaling—with the current data volume, we have

| Model size | Patch size | | | | |
|---|---|---|---|---|---|
| | 16×16 | 8×8 | 4×4 | 2×2 | 1×1 |
| *with 128×128 resolution inputs:* | | | | | |
| Tiny | 72.6 | 78.5 | 80.4 | 80.6 | 80.7 |
| Small | 77.6 | 80.5 | 80.9 | 81.2 | 81.4 |
| Base | 79.0 | 81.5 | 81.8 | 82.2 | 82.4 |
| Large | 79.8 | 82.2 | 82.6 | 82.9 | 83.1 |
| *with 224×224 resolution input:* | | | | | |
| Tiny | 78.2 | 80.9 | 81.3 | 81.7 | 81.9 |
| Small | 81.8 | 83.0 | 83.5 | 83.7 | 83.8 |
| Base | 82.6 | 83.9 | 84.3 | 84.5 | 84.6 |

*Table 4.* **Scaling both patch and model sizes.** The gains from patch size scaling and model size scaling are not conflicting; combining both can lead to further performance improvements. The numbers denote ImageNet accuracy (%) with Adventurer models. We associate the results with different shades for clear observation.

| Model | Length | By extending | By scaling |
|---|---|---|---|
| DeiT-Base, 128×128 input | 64 | 78.1 | 78.1 |
| | 256 | 78.2 (+0.1) | 81.0 (+2.9) |
| | 1,024 | 78.2 (+0.1) | 82.3 (+4.2) |
| Adventurer-Base, 224×224 input | 196 | 82.6 | 82.6 |
| | 784 | 82.7 (+0.1) | 83.9 (+1.3) |
| | 3,136 | 82.8 (+0.2) | 84.3 (+1.7) |
| | 12,544 | 82.8 (+0.2) | 84.5 (+1.9) |

*Table 5.* **Ablation of sequence length.** Extending the sequence length alone does not yield significant improvements (column "by extending"), whereas reducing patch size and lowering information compression rate is the primary source of performance gains (column "by scaling"). Performance is measured by ImageNet-1k accuracy (%), with longest sequences highlighted in blue.

| Patch | Length | Memory (per image) | GPU hours | |
|---|---|---|---|---|
| | | | DeiT-Base | Adv-Base |
| 16 | 196 | 62MB | 0.36 | 0.45 |
| 8 | 784 | 252MB | 1.86 | 1.76 |
| 4 | 3,136 | 1,024MB | 9.79 | 6.86 |
| 2 | 12,544 | 4,057MB | 80.06 | 27.45 |
| 1 | 50,176 | 16,118MB | 967.99 | 115.08 |

*Table 6.* **Computational overhead** for training a DeiT-Base and Adventurer-Base at 224×224 resolution inputs and different patch sizes. Memory usage is calculated based on the per-image consumption in ViT. GPU hours (for each ImageNet epoch) are estimated on a single A100 GPU. The models are trained at Float16 precision with FlashAttention (Dao et al., 2022) applied in ViT. The detailed evaluation protocol can be found in Appendix.

that extends the input sequence interpolating on existing tokens. Specifically, in this comparison, we retain the original large patch size (16×16) but perform spatial interpolation on the tokens produced by the patch embedding, by which we extend the input sequence without introducing any new information. As shown in Table 5, this approach does not bring substantial improvements to the model's performance (see column "By extending"). In contrast to the significant gains achieved through patchification scaling (*e.g.*, 4.2% accuracy on ImageNet), this ablation study effectively demonstrates that the benefits of our approach primarily stem from unlocking the visual information compressed by large patch sizes, enabling the model to focus on more detailed visual features, while simply scaling the sequence length itself has only a minimal impact on performance.

### 4.4. Discussions

We summarize the computational requirements involved in the patchification scaling experiments in Table 6. As shown, the super-long visual token sequences associated with small patch sizes impose a significant hardware overhead on ViT architectures. This overhead was indeed a major challenge around five years ago, when V100 GPUs with 16/32GB memory remained the mainstream hardware for AI training. However, with rapid advancements in hardware development, efficient parallel computing mechanisms, as well as low-complexity visual architectures, the idea of ***"learning from pixels"*** has become increasingly feasible.

In the experiments, we have demonstrated many key benefits of patchification scaling, such as direct performance improvements, reduced dependence on decoders, and the ability to overcome many limitations of parameter scaling and input size scaling. These emerging properties suggest that, when computational resources allow, we should gradually reduce or even abandon the spatial compression mechanisms in vision encoders, fully exploiting all the information inherently provided by the data. We hope this paper can pro-

already observed the limitations of parameter scaling in vision models (Zhai et al., 2022; Dehghani et al., 2023), while it is promising to see more future breakthroughs in visual encoding with the help of this new scaling dimension.

**Impact of sequence length.** Intuitively, scaling down the patch size has two direct effects. First, smaller patch sizes allow the model to receive richer, more fine-grained input information, which can greatly benefit its inference abilities. Second, reducing the patch size directly extends the token sequence, and for token mixers like self-attention or Mamba, longer sequences inherently expand the model's representational space, enhancing its capabilities in feature processing. Both factors can potentially have a significant impact. We seek to demonstrate that the performance improvement from reducing the patch size primarily arises from the information gain due to a lower compression rate, rather than from the enhanced representational capacity associated with an extended sequence length.

We conduct a direct ablation study: in contrast to our patch size scaling approach, we set up an additional experiment

vide insights and inspiration for a transition from the current patch-based compressive encoding paradigm to pixel-based non-compressive visual foundation models.

## 5. Conclusion

In this work, we conduct extensive studies in reducing the spatial compression rate in patchification layers and discover a new scaling dimension for visual encoding, which we term **Patchification Scaling Laws**. The new scaling laws suggest that, with more computational resources invested, leveraging smaller patch sizes consistently leads to improved predictive performance. This conclusion is broadly applicable across various vision tasks, different input resolutions, and diverse model architectures. As a by-product, we also identify an interesting emerging property of patchification scaling: when the encoder patch size becomes sufficiently small, the benefits provided by task-specific decoder heads diminish significantly. We hope the discoveries in this paper can provide a solid theoretical foundation for the future pixel learning paradigm and the development of non-compressive visual foundation models.

## Acknowledgments

This work is supported by ONR: N00014-21-1-2690 and National Eye Institute (NEI) with Award ID: R01EY037193. We would like to thank TPU Research Cloud (TRC) program and Google Cloud Research Credits program for partially supporting our computing needs.

## Impact Statement

This paper presents work whose goal is to advance the field of Machine Learning. Our experiments involved approximately 50,000 A100 GPU hours, which is considered a modest level of resource consumption compared to large-scale vision or language model research. While there are many potential societal consequences of our work, none are significant enough to warrant specific highlighting in this context. We believe the ethical impacts and societal implications are well-aligned with the advancement of machine learning technology.

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

# Appendix

## A. More Technical Details

**The detailed configuration** of the models used in this paper are elaborated in Table 7. For ViTs, We basically follow the configurations in the DeiT series models (Touvron et al., 2021a; 2022b), but change the default patch size of DeiT-Huge to 16×16, the same as the other DeiT models. For Adventurer, we scale up the model to a huge size following the same rule of DeiT; we set its embedding dimension to 1,280, keeping its original MLP ratio and employ 32 blocks in total.

| Model | Embedding dimension | MLP dimension | Blocks | Parameters |
|---|---|---|---|---|
| DeiT-Tiny (Touvron et al., 2021a) | 192 | 768 | 12 | 5M |
| DeiT-Small (Touvron et al., 2021a) | 384 | 1,536 | 12 | 22M |
| DeiT-Base (Touvron et al., 2021a) | 768 | 3,072 | 12 | 86M |
| DeiT-Large (Touvron et al., 2022b) | 1,024 | 4,096 | 24 | 304M |
| DeiT-Huge (Touvron et al., 2022b) | 1,280 | 5,120 | 32 | 631M |
| Adventurer-Tiny (Wang et al., 2024c) | 256 | 640 | 12 | 12M |
| Adventurer-Small (Wang et al., 2024c) | 512 | 1,280 | 12 | 44M |
| Adventurer-Base (Wang et al., 2024c) | 768 | 1,920 | 12 | 99M |
| Adventurer-Large (Wang et al., 2024c) | 1,024 | 2,560 | 24 | 346M |
| Adventurer-Huge (Wang et al., 2024c) | 1,280 | 3,200 | 32 | 759M |

*Table 7.* **Model configurations.** All models have a 16×16 patch size by default.

**Protocols of estimating memory and GPU hours.** In Table 6, we present an estimation of the GPU memory and training hours required for DeiT and Adventurer. Here we give more details of how they are evaluated. We calculate the memory consumption by each image. That means, the reported numbers have excluded the memory used for storing the model, optimizer, and other hyper-parameters. The actual memory demand increases linearly with batch size. To evaluate the GPU hours required for training, we set a total batch size of 1,024 and use the minimum number of nodes necessary for training (depends on the total memory demand). Each node is equipped with 8 A100/80GB GPUs. The estimated training hours are then multiplied by the total number of GPUs used to ensure that the reported numbers are normalized.

| Config | Tiny/Small/Base | Large/Huge |
|---|---|---|
| optimizer | AdamW | |
| base learning rate | 5e-4 | 2e-4 |
| weight decay | 0.05 | 0.3 |
| epochs | 300 | 200 |
| optimizer betas | 0.9, 0.999 | 0.9, 0.95 |
| batch size | 1024 | 4096 |
| warmup epochs | 5 | 20 |
| stochastic depth (drop path) | 0.1 | 0.2 |
| layer-wise lr decay | ✗ | |
| label smoothing | ✗ | |
| random erasing | ✗ | |
| Rand Augmentation | ✗ | |
| repeated augmentation | ✓ | |
| ThreeAugmentation | ✓ | |

*Table 8.* **Recipe of the pretraining stage,** for 64×64 or 128×128 pixel inputs.

**Training recipes.** In this work, we train DeiT-Tiny, Small, and Base with the official repository (Touvron et al., 2021a) and recipe. For DeiT-Large and Huge, there is not training configuration in the original DeiT paper so we follow the supervised training pipeline reported in (He et al., 2022). Note that the Pixel Transformer (Nguyen et al., 2024) which conduct pixel tokenization experiments with low resolution images (28×28) employs the same training recipe.

For Adventurer, we mostly follow its original multi-stage strategy (Wang et al., 2024c) to train our models. Specifically, for 64×64 resolution inputs, we simply perform the pretraining stage (shown in Table 8) for 300 epochs for all model sizes. For 128×128 resolution inputs, we additionally perform a finetuning stage (shown in Table 9) for enhanced results. For the standard 224×224 resolution inputs, we follow the practice of Mamba-Reg (Wang et al., 2024b) and Adventurer (Wang

et al., 2024c) to load the pretrained model at 128×128, performing an intermediate training stage (shown in Table 10) for 100 epochs and then a finetuning stage for 20 epochs. We highlight that this multi-stage training strategy is highly efficient for our experiments as we can fully exploit the models pretrained at lower resolutions. For example, we only need to train the 224×224-input models for 120 epochs since we can load the weights pretrained at 128×128 resolution inputs.

Notably, for both DeiT and Adventurer, *there is no need to adjust training recipes for different patch sizes*, which we consider to be one of the flexible and practical advantages of patchification scaling.

| Config | Small/Base | Large |
|---|---|---|
| optimizer | AdamW | |
| base learning rate | 1e-5 | 2e-5 |
| weight decay | 0.1 | 0.1 |
| epochs | 20 | 50 |
| optimizer betas | 0.9, 0.999 | 0.9, 0.95 |
| batch size | 512 | 512 |
| warmup epochs | 5 | 5 |
| stochastic depth (drop path) | 0.4 (S), 0.6 (B) | 0.6 |
| layer-wise lr decay | ✗ | 0.95 |
| label smoothing | 0.1 | |
| random erasing | ✗ | |
| Rand Augmentation | rand-m9-mstd0.5-inc1 | |
| repeated augmentation | ✗ | |
| ThreeAugmentation | ✗ | |

*Table 9.* **Recipe of the finetuning stage,** for 128×128 or 224×224 pixel inputs.

| Config | Small/Base | Large |
|---|---|---|
| optimizer | AdamW | |
| base learning rate | 5e-4 | 8e-4 |
| weight decay | 0.05 | 0.3 |
| epochs | 100 | 50 |
| optimizer betas | 0.9, 0.999 | 0.9, 0.95 |
| batch size | 1024 | 4096 |
| warmup epochs | 5 | 20 |
| stochastic depth (drop path) | 0.2 (S), 0.4 (B) | 0.4 |
| layer-wise lr decay | ✗ | 0.9 |
| label smoothing | ✗ | |
| random erasing | ✗ | |
| Rand Augmentation | ✗ | |
| repeated augmentation | ✓ | |
| ThreeAugmentation | ✓ | |

*Table 10.* **Recipe of the intermediate training stage,** for 224×224 pixel inputs.

