# OpenReview forum: "Scaling Laws in Patchification: An Image Is Worth 50,176 Tokens And More"
_ICML.cc/2025/Conference — ICML 2025 poster_

### Official Review · Reviewer_UjKW · 2025-02-15

**Overall Recommendation:** 3

**Summary:**

This paper aims to explore how the performance of vision transformers changes when the patch sizes are scaled up. The main contribution is the scaling law, which has not been presented in previous work, that as the patch sizes get smaller, the classification and segmentation performance can be lifted. The authors also show that the benefit of scaling up patch number is larger than scaling up the model size. This is really an interesting phenomenon.

**Claims And Evidence:**

The authors have made extensive experiments to support the claims.

**Essential References Not Discussed:**

See the 'Experimental Designs Or Analyses' section.

**Experimental Designs Or Analyses:**

The experimental designs are thoroughout. The authors conduct a series of experiments to demonstrate the claims and the designs are also meaningful.

- One concern about the experimental design is that the authors only use two baseline models, i.e., DeiT and Adventurer. I would like to see whether the scaling law still exists when stronger baselines are used, i.e., [A] and [B].

- In addition, the experimental designs are all based on the ViT-like plain architectures. This type of architectures has been widely used in vision tasks, but the pyramid architecture is also important. For example, Swin Transformer is such a type of architecture adopting a pyramid network. Have the authors conducted experiments on this type of architectures?

[A] All tokens matter: Token labeling for training better vision transformers. NeurIPS, 2021.
[B] Early convolutions help transformers see better. NeurIPS, 2021.

**Methods And Evaluation Criteria:**

The criteria is widely-used.

**Other Comments Or Suggestions:**

Actually, when I read the introduction section of this paper, I was supposed that a novel method is presented to solve the computation overhead when using smaller patches. If such a method is proposed, I think the overall quality of this paper could  be further improved.

**Other Strengths And Weaknesses:**

Strengths:

- The presentation of this paper is good. The authors clearly explain the motivation of this paper and the method of this paper is easy to follow.
- Though the method does not present an interesting method, the phenomenon observed by the authors are interesting.

Weakness:

- The paper [B] has shown that adding early convolutions to the plain ViT model can largely improve the model performance. However, the authors did not compare with this type of works, which aim to replace the patchification methods. It would be good to add some comparisons about this.

- In Fig. 3, it is good to see the comparison between patch size scaling and parameter scaling. However, it is straightforward to come up with a new question: Could these two types of scaling methods benefit from each other? In other words, could the performance be further improved if fine-grained patchification is used when taking higher-resolution images as inputs?

- The authors claim that the proposed approach is also applicable to Mamba-like models. However, it seems that there are no experimental results supporting this.

[B] Early convolutions help transformers see better. NeurIPS, 2021.

**Questions For Authors:**

Not applicable.

**Relation To Broader Scientific Literature:**

The scaling law presented in this paper has not been discussed in previous work to my knowledge.

**Theoretical Claims:**

These is no theoretical claims. Not applicable.

---

> ### Author Rebuttal · Authors · 2025-04-01
>
> We sincerely thank the reviewer for their careful evaluation and thoughtful comments. Detailed responses can be found below.
>
> **Q1: *Experiments with other architectures (e.g., Swin Transformer, LV-ViT)***
>
> Thank you for your insightful feedback. We would like to highlight that our conclusion also holds for pyramid networks such as Swin Transformer (see results in the response of Q1 for Reviewer DwN7) and stronger ViT baselines such as LV-ViT (see table below). To complete this experiment within a limited time frame, here we use 112×112 input images and employ LV-ViT-S as the backbone. Under this stronger baseline, we observe a performance trend similar to that in the standard ViTs, which demonstrates the architecture-wise robustness of patch size scaling. We will include the corresponding discussion in the revised version.
>
> | Patch | 16x16 | 8x8 | 4x4 | 2x2 |
> |-------|-------|-----|-----|-----|
> | Acc.  | 78.0  | 81.1| 82.5| 83.2|
>
> **Q2: *Comparison with early convolutions to the plain ViT model.***
>
> Thank you for your suggestion of comparing with early convolutions. We summarize the results of patchification scaling with early convolution in the table below. As shown, when the patch size is at the standard 16×16 level, applying early convolution brings a noticeable performance gain. However, this benefit diminishes rapidly as the patch size decreases, and becomes negligible at 8×8. In simple terms, the basic idea behind early convolution is to decompose a single 16×16 kernel size and 16×16 stride convolution layer into a stack of smaller kernels (e.g., 3×3) with smaller strides (e.g., 1×1 or 2×2). This approach effectively addresses the training instability caused by the abrupt spatial resolution drop at the patchification layer in standard ViTs. However, as the patch size decreases, this issue becomes much less pronounced. We are encouraged to see that our paper shares some core insights with this line of research—namely, that large-kernel patchification in standard ViTs can limit model expressivity. The difference is that we take a more direct approach to reduce the spatial downsampling effect in patchification. When extended to the extreme case of pixel tokenization, this issue is already fundamentally resolved. We will include these results and discussions in the revised version.
>
> | Patch size | Original accuracy | Early conv. accuracy |
> |------------|-------------------|----------------------|
> | 16x16      | 82.6              | 83.1                 |
> | 8x8        | 83.9              | 83.9                 |
>
> **Q3: *Could these two types of scaling methods (patch size and parameter size) benefit from each other?***
>
> Thank you for your insightful question. Yes, they do benefit each other and we have related results in Table 4 and discussions in line 364-376r. Table 4 shows a consistent upward trend from the top-left (small model and large patch size) to the bottom-right (large model and small patch size) corner. Additionally, as observed from both Table 1 and Table 4, reducing the patch size and increasing the input resolution can both have a positive impact, demonstrating the good potential of jointly employing these scaling dimensions.
>
> **Q4: *Application to Mamba-like models.***
>
> Thank you for your feedback. We evaluate the patch size scaling performance with ViT and Adventurer models, where for Adventurer, we are actually using its Mamba-based setup so the application to Mamba models is already included. We will clarify this point in the revision.
>
> **Q5: *I was supposed that a novel method is presented to solve the computation overhead when using smaller patches. If such a method is proposed, I think the overall quality of this paper could be further improved.***
>
> We appreciate your constructive comment on this point! In fact, we first identified a solution to address the computational challenges brought by small patch sizes before conducting the patchification scaling study: we chose to carry out the main experiments using the Adventurer model, whose computational cost scales linearly—as opposed to ViT’s quadratic scaling—with respect to sequence length. This linear complexity fundamentally resolves the computation bottleneck, which enables us to perform pixel tokenization experiments using modest computational resources (256 A100 GPUs). We also discuss the advantages of this linear architecture in Table 6: in the most demanding training setup with the longest sequences, Adventurer achieves an 8.4× speedup compared to ViT with FlashAttention. This substantial efficiency gain is what made our large-scale pixel tokenization experiments practically feasible.

---

> > ### Comment · Reviewer_UjKW · 2025-04-09
> >
> > Thanks for the responses. My concerns have been solved. Other reviewers have difference concerns on this paper, but I think the observation of this paper is interesting. I would like to keep my rating unchanged.

---

### Official Review · Reviewer_KY13 · 2025-03-15

**Overall Recommendation:** 2

**Summary:**

The authors perform a study regarding the size of patches used in modern vision transformers or state space models. Utilizing the Adventurer state space model, the authors are able to experiment with resulting sequence lengths of up to 50,176 tokens.
The work arrives at the conclusion that there exists a scaling law that holds until each input pixel is represented by a dedicated patch/token. Smaller patches naturally align well with dense prediction tasks like semantic segmentation, where the need for a decoder diminishes with the use of smaller patches.

## update after rebuttal

I thank the authors for their efforts in answering my questions and providing additional experiments. Still, I am of the opinion that the introduction would benefit from references to other works, that already experimented and observed the benefit of smaller patches.
Furthermore, the authors explain the improvement by a reduction of the information loss in the patchification step. Which is very reasonable and certainly true for smaller models. The insightful experiment presented in Figure 4. shows that the scaling potential is bounded by the original image resolution. I think a similar experiment with respect to the hidden dimension of the model would be of high relevance.  Thus, I still think that the paper would greatly benefit from an experiment that investigates the scaling potential in a regime where no compression is needed to create patches.

The experiments provided during the rebuttal support the information loss argument, as the smaller patch versions consistently show a much smaller reconstruction loss. Yet on a second glance, the results are not totally plausible, as they show no impact of the bottleneck dimension of the autoencoder whatsoever.

**Claims And Evidence:**

The work claims that there exists a scaling law when it comes to the size of patch tokens used in transformer or state space models for image processing. The authors provide empirical results that clearly support the claim.

**Essential References Not Discussed:**

Not essential, but there are quite few works in self-supervised learning that already utilize smaller patches to improve performance. And especially for smaller models that is a widely used practice. Already the 14x14 ViT Huge model in the ViT paper suggests that the potential benefit of smaller patches has been known for quite some time.
E.g. [1-3], especially in 3 the authors already work with small 4x4 patches to achieve an advantage in the resulting per-parameter comparison to other models

[1] Caron, Mathilde, et al. "Emerging properties in self-supervised vision transformers." Proceedings of the IEEE/CVF international conference on computer vision. 2021.
[2] Zhou, Jinghao, et al. "Image BERT Pre-training with Online Tokenizer." International Conference on Learning Representations.
[3] Assran, Mahmoud, et al. "Masked siamese networks for label-efficient learning." European conference on computer vision. Cham: Springer Nature Switzerland, 2022.

**Ethical Review Concerns:**

None.

**Experimental Designs Or Analyses:**

Yes. Experiments follow recipes proposed in the respective publications (e.g. DeiT and Adventurer). Adaptations are listed in the Appendix.

**Methods And Evaluation Criteria:**

Supervised training on ImageNet is a standard evaluation that allows for a comparison to a wide range of models. Semantic segmentation on ADE20k is common as well. The authors clearly state that they focus on decoder free evaluation as it aligns with the benefits of smaller patches.

**Other Comments Or Suggestions:**

typo fig 3b. form - from

**Other Strengths And Weaknesses:**

Strengths:
- the work provides empirical evidence that smaller and smaller patches steadily improve the performance of visual backbones and follow a scaling law.
- both for transformer and state space models.
- the experimental setup is well documented.
- Additional studies that one patch per pixel is the optimal size and that more patches than pixels shows no benefits.

Weakness:
- no technical or theoretical contribution
- the benefit of smaller patches has been known and has been exploited for quite some time

**Questions For Authors:**

Concerning the argument that the scaling law can be attributed to the information loss in the patch creation step . Why should this hold in the case where the hidden dimensions are large enough to simply stack the corresponding pixels?

**Relation To Broader Scientific Literature:**

The contribution of this paper adds another example to the set of scaling laws that have shown empirically that an exponential increase in resources leads to a linear performance improvement.

**Theoretical Claims:**

No theoretical claims.

---

> ### Author Rebuttal · Authors · 2025-04-01
>
> We sincerely appreciate your constructive review, insightful feedback, and recognition of the empirical contribution of this work. The detailed response is summarized below.
>
> **Q1: *No technical or theoretical contribution.***
>
> Thanks for the comment. As you noted, this is indeed an experiment-driven study, and our focus is on delivering empirical results to demonstrate a new scaling law. We respectfully ask the reviewers to take into consideration that all studies on scaling laws are inherently grounded in empirical observations. For large vision or language models, it is extremely difficult—if not impossible—to theoretically derive precise upper or lower bounds on their fitting capacity. Therefore, empirical evidence remains the most practical and informative approach for uncovering such trends.
>
> If we consider the initial scaling law proposed by Kaplan et al. (2020) as a theoretical foundation, this work will be a meaningful extension of their theories in vision. Specifically, in the initial study, they show that in language modeling tasks, when the model size (parameter count $M$) and the amount of training data (token count $N$) increase, the perplexity (or loss) on the validation set tends to follow a near power-law relationship:
>
> $ L(N,M)\approx\alpha N^{-\beta}+\gamma M^{-\delta}+\text{(smaller interaction terms)}$
>
> where $L$ represents the validation loss (e.g., cross-entropy), and
> $\alpha,\beta,\gamma,\delta$ are constants fitted from large-scale experiments. In this work, we not only reproduced similar scaling trends on vision tasks, but also validated the second component of the scaling law (the token count) in the vision domain. Specifically, we show that increasing token count in vision can be achieved by reducing patch size rather than solely increasing dataset size, which we believe is an important theoretical extension of the original scaling law foundation. We sincerely appreciate your suggestion on this point and will include the corresponding discussion in the revised version.
>
> **Q2: *The benefit of smaller patches has been known and has been exploited for quite some time***
>
> Thanks. We agree that many existing studies have observed that using smaller patch sizes within a certain range can improve prediction performance. However, prior to our work, these observations remained scattered and lacked a unified theoretical or empirical framework. In contrast, our study elevates patchification scaling from isolated findings to a law-level conclusion.
>
> We believe this distinction is substantial—not only in terms of the conclusions themselves, but also in their implications for guiding future progress in the community. For example, before the NLP scaling laws were formally introduced by Kaplan et al., practitioners had already noticed from experience that "larger models and more data tend to yield lower loss." Yet such heuristic insights were insufficient to offer reliable guidance or theoretical grounding for large-scale language model development.
>
> The formulation of scaling laws provided the community with a much clearer signal: the scaling curve had no evident upper bound, or at least we were far from reaching it. This encouraged researchers to confidently invest significant resources in scaling up language models, ultimately contributing to the breakthrough success of models like the GPT series.
>
> In the same spirit, we would like to highlight the contribution of our work. We aim to offer a long-term, principled perspective on vision model development. The value of patchification scaling laws lies in showing that patch size is a reliable scaling dimension—even at today’s typical input scales, shrinking the patch size down to 1×1 continues to yield noticeable gains. This suggests that for those seeking to push the limits of model performance, reducing patch size remains a viable and promising direction.
>
> **Q3: *Concerning the argument that the scaling law can be attributed to the information loss in the patch creation step . Why should this hold in the case where the hidden dimensions are large enough to simply stack the corresponding pixels?***
>
> When the hidden dimension exceeds the total number of pixels within a patch, the patch embedding may theoretically support a lossless projection. However, in practice, the high-dimensional hidden features are optimized to represent the holistic semantic of the whole patch, rather than preserving the pixel-level representations. This is because the token mixing process (e.g., self-attention) is performed on patch-level features. That says, the different segments within a feature vector undergo the same operation in token mixing. Moreover, the hidden dimension cannot be flexibly scaled up since it quadratically increases parameter count for all linear projection layers and can easily lead to training collapse (e.g., also observed in Dehghani, et al., 2023).
>
> **Q4: *Typo fig 3b.***
>
> Thanks. We will fix it in the revision.

---

> > ### Comment · Reviewer_KY13 · 2025-04-03
> >
> > Thank you for answering my questions.
> >
> > @W1: When comparing this work and Kaplan 2020, I think there is still a difference regarding novelty/superprise factor of the results and scale/scope of the recpective experiments in favor of Kaplan 2020.
> >
> > @W2: Was not solely about the distinction of your work, but also about the fact that your work does not mention any of these observations.
> >
> > @Q1: If I read your answer corrrectly, you as well attribute the improved performance of models that use smaller patches to the increased capacity due to an exponential increase in neurons, which becomes even more relevant when the level of abstraction and sparseness increases. Combined with the fact that attention and feed forward layers use weight sharing, no additional parameters are introduced  and overfitting/ training instabilities are less of a problem. Which is not exactly  a preservation of information.

---

> > > ### Author Response · Authors · 2025-04-04
> > >
> > > We appreciate your detailed response to our rebuttal.
> > >
> > > **@W1:** We agree that Kaplan 2020, as the first work to propose scaling laws, undoubtedly has profound novelty and practical impact. However, our work fills the gap in previous visual scaling laws and introduces a new dimension orthogonal to parameter scaling, which provides valuable guidance for the future development of visual models.
> > >
> > > **@W2:** In the paper, we discussed some prior observations that using smaller patch sizes can help improve performance. For example, in the Introduction, we note that previously, reducing DeiT's patch size to 8x8 resulted in significant performance gains (lines 28-31r), and Nguyen 2024 observed substantial gains using pixel patchification on small input images at a 28x28 resolution. We want to again highlight that these observations are sporadic, and our work systematically studies this issue and concludes it into a scaling law. In the revised version, we will provide a detailed summary of previous work on the impacts of patch size discoveries.
> > >
> > > **@Q1:** Regarding the information loss and preservation problem, here we would like to give direct evidence that large patches easily lead to an information loss and smaller patch sizes are effective solutions. We conduct a simple pixel-reconstruction experiment to verify whether information can be completely restored after patchification. We employ a three-layer model consisting of a patchification layer (i.e., a $p\times p$ kernel and stride convolution), layer normalization, and a de-patchification layer (another similar convolutional layer that brings patches back to pixel dimensions). We then train this model on ImageNet using a pixel-to-pixel L2 loss, with a batch size of 64 for 60k iterations. The results are shown in the following table. We observe that the reconstruction loss is closely related to the patch size but largely independent of the hidden dimension. At a 16x16 patch size, we see a significant reconstruction loss, e.g., 0.211 for the base-sized hidden dimension (768). However, when the patch size is reduced to 1x1, the loss dramatically decreases to 0.012, indicating that we fundamentally solve the issue of information loss in this process.
> > >
> > > | Hidden dimension | Patch size ($p$) | Reconstruction loss ($\downarrow$) |
> > > |------------------|------------------|---------------------|
> > > | 384              | 16               | 0.204               |
> > > | 384              | 4                | 0.068               |
> > > | 384              | 1                | 0.015               |
> > > | 768              | 16               | 0.211               |
> > > | 768              | 4                | 0.065               |
> > > | 768              | 1                | 0.012               |
> > > | 1280             | 16               | 0.202               |
> > > | 1280             | 4                | 0.060               |
> > > | 1280             | 1                | 0.014               |
> > >
> > > It's noteworthy that theoretically, as long as the hidden dimension in the linear layer is sufficiently large, it can preserve all the pixel information within the original patch. However, in practice, any passage through a non-linear layer (such as LayerNorm) compresses these fine-grain details. Thus, as observed in this experimental result, increasing the hidden dimension does not mitigate information loss. In both ViT and Adventurer, each token mixer is preceded by a LayerNorm, resulting in significant information loss with the traditional 16x16 patch size.

---

### Official Review · Reviewer_vQ14 · 2025-03-15

**Overall Recommendation:** 2

**Summary:**

The paper evaluates the performance (test loss) of vision models (standard ViT and Adventurer) against different patch sizes. The paper's findings are that with reduced patch size, the performance of the networks increases and in the extreme case of 1x1 patch size, the segmentation tasks do not need a decoder head. The paper evaluates the patchification scaling on the standard ImageNet-
1k classification, ADE20k semantic segmentation, COCO object detection and instance segmentation benchmarks.

**Claims And Evidence:**

1. The paper examines how compressive encoding affects visual representations and whether patch size can be a new scaling dimension for modern visual architectures.
    It provides evidence in the form of empirical results obtained by changing the patch size down to 1x1 (pixel level).
2. The paper claims tokenization/patchification is another dimension for scaling law. The evidence is provided by reducing the patch and getting marginal gains.

**Essential References Not Discussed:**

It's surprising that "On the relationship between self-attention and convolutional layers" by JB Cordonnier et. al., which was the basis of ViTs and started with the patch size of 2x2 (which was later changed to 16x16 in ViT paper) has not been discussed at all.
Comrpession and compression related topics (e.g. information theory) are entirely missing form the paper.

**Experimental Designs Or Analyses:**

The experimental design and ablations in the paper are either flawed or entirely absent. Please refer to the other sections for the stated problems.

**Methods And Evaluation Criteria:**

This study should have been done very carefully and meticulously because there are many aspects and variables that need to be carefully analyzed and ablated.
The overall method simply involves getting two networks and reducing the patch size down to 1x1 (not for pure transformers) and then showing results on vision and timeseries tasks.
The issue is that the method is counter to common wisdom of balancing the compute and gains trade off. It does not make sense to increase compute just because more hardware is available without getting real gains from it. The method showcases its results on metrics without ever indulging into the GLOPs, test and training time compute, throughput and other aspects that come with reducing the patch size of the input.

The paper raises the slogan of “a pixel is worth a token” (line 233) but does not practice it in ViT because the “a pixel is worth a token” requires a lot of resources.

Line 312-313L: The paper provides misleading sense  of great results but the comparison is unfair since the numbers are compared to 32x32 patch size instead of the mainstream/common 16x16 or 14x14 patch size. The wosre results on uncommon 32x32 patch size makes it looks like the results are good (which they arent)

**Other Comments Or Suggestions:**

Missing footer. Not using the ICML template or the template is altered.

Line 250-257: Sounds like the paper is claiming the credit for the method "Adventurer" only for increasing the input token size by reducing the patch size. Increasing the input size shouldn't be posed as an achievement.

Line 351R: The same condition does not hold for the paper's own method.

**Other Strengths And Weaknesses:**

Weaknesses:
1. The overall idea of scaling is to increase the generalization capability of the models. It is not supported by the evidence shown in the paper that patchification leads to better generalization. Moreover, smaller patchification leads to more computation without having significant benefits.
2. Analogous patchification in NLP would be the character-level patchification which has been shown not to be optimal. It is not clear why and why not similar will be true for vision models from this work.
3. There are claims and arguments (discussed above) that are not supported by any experiments or ablation studies.
4. There is no effort to introduce or quantify the information loss lost by compressive encoding from the information theory perspective.
5. There is no ablation that discusses the patchification vs. embedding dimension size trade-off.
6. The results themselves are only marginally better and yet underperforming to SOTA methods with a greater patch size (Spatial Mamba for example has better results).

**Questions For Authors:**

Line 15-16R (Right paragraph): "We  argue  that  this  operation  often  incurs irreversible information loss to visual inputs." There is no reference, ablation or supporting evidence provided for this argument.

Line 105: Why exactly computation needs to be scaled up?

Line 114-115: Single pixel patch (or even few pixels patch) expanded to embedding dimension is not even compressive paradigm anymore but "expansive" regime so to say. There is no discussion on compression ratio.

Line 324R: "patch size scaling not only exhibits a better computation-accuracy tradeoff": Where is the trade-off provided?

**Relation To Broader Scientific Literature:**

NA

**Theoretical Claims:**

This paper actually could ground itself from a mathematical point of view and framework when talking about compression and information contained in the embedding vectors of a given size. The lack of such framework and relying entirely on empirical results is not much convincing.

---

> ### Author Rebuttal · Authors · 2025-04-01
>
> We appreciate your comments and feedback during the review stage. We‘d like to respectfully point out that your review contains several factual misunderstandings of our paper. These points were clearly presented in the main text, and we'd like to first clarify them below:
>
> 1. ***Absence of 1x1 patch size for ViT:*** This is factually incorrect. We do implement pixel patchification in ViTs, albeit at lower input resolutions. The results are clearly shown in the first row of Tab.1.
>
> 2. ***“The overall method… showing results on vision and timeseries tasks.”*** Our work is solely focused on vision tasks, and there is no mention or evaluation on time series data anywhere in the paper.
>
> 3. ***“without ever indulging into the GLOPs, test and training time compute”*** This is completely inaccurate. We provide thorough analysis of FLOPs (Fig.3), memory usage and training time (Tab.6), and discuss computational costs of patch scaling in multiple places (e.g., Lines 324–329, 436l–408r).
>
> 4. ***"no ablation that discusses the patchification vs. embedding dimension size trade-off."*** We compared patch size and parameter size scaling in the first experiment of the ablation study in Section 4.3. The embedding dimension is highly equivalent to the parameter size. E.g., in the standard scaling of ViT models from ViT-T to ViT-B, the only change lies in the embedding dimension; from ViT-B to larger models such as ViT-L, it still involves increased embedding dim while depth is also changed.
>
> **Q1: *Compare to 32 patch size***
>
> Our study focuses on exploring scaling laws of patchification, which means we need to examine both scaling up and scaling down. We aim to collect a wide-range performance curve to demonstrate that the gains are smooth and consistent across different patch sizes. The 32 patch size results are reported to provide a more comprehensive view; we did not hide the results for the mainstream 16 patch size—they are shown in the same table and scaling from it leads to a +2.2 box AP.
>
> **Q2: *Generalization capability***
>
> We demonstrated that patch scaling consistently leads to improved predictive performance, which is generalizable for different models, tasks, and input resolutions. Below we include more results of generalization capability against out-of-domain data, where our conclusions hold true for ImageNet variant test samples as well.
>
> |Patch|IN-v2|IN-R|IN-A|IN-S|
> |-|-|-|-|-|
> |16|71.7|87.5|31.4|36.8|
> |8|72.9|87.9|36.6|38.3|
> |1|74.3|88.8|41.6|39.5|
>
> **Q3: *Patchify in NLP***
>
> Word-level tokenization has indeed been prominent in NLP. However, we believe this is only a phase-based conclusion but no one has ever proven that it is the only correct approach. In contrast, recent studies(e.g. Byte Latent Transformer) have shown that byte-level LLMs can outperform word-level models and offer better scaling potential. Thus, whether patchify is necessary remains an open question that deserves further investigation. Moreover, directly transferring conclusions from NLP to CV is not always appropriate. Images and text differ significantly in terms of information density; a pixel cannot be naively equated to a character in text. These differences call for careful study, and our work aims to contribute to this ongoing exploration.
>
> **Q4: *Quantify the information loss from the information theory perspective.***
>
> This paper is not a theory-oriented study, but rather an empirical investigation into scaling laws. We believe both empirical and theoretical perspectives are equally important, and our experimental results can serve as a solid foundation for future theoretical work. In fact, our evaluations already offer explanations from an information point of view, in which compression can be assessed via entropy differences. Fig.1 explicitly shows how cross-entropy loss changes during patch scaling, which is actually measuring the KL divergence between ground-truth distribution and encoded distribution. This typically serves as a direct proxy for compression in information theories.
>
> **Q5: *Results marginally better; underperform SOTA models***
>
> Our focus and contribution is figuring out how patch size affects the performance of vision models, rather than chasing SOTA on open-ended architectures. Whether a performance gain is considered marginal is subjective and varies by context. For fixed architecture, improving accuracy from 82.6 to 84.6 is a significant achievement. It is widely acknowledged that hierarchical models (e.g., Swin, SpatialMamba) are more capable of fitting ImageNet-scale datasets compared to plain models, but plain models have wider applications in multimodal tasks. We focus on patch scaling within plain architectures so the performance gap with SOTA models is expected. In our response to R#DwN7 we also show that patchification scaling laws hold true for Swin, so we believe that future follow-up work applying patch size scaling to SOTA architectures could reasonably expect further performance improvements.

---

> > ### Comment · Reviewer_vQ14 · 2025-04-08
> >
> > Thank you for your rebuttal. Ironic that the mistakes I have made while writing the review have forced me to write the rebuttal on my own review. I apologize for the mistakes.
> > 1. Clarification on the absence of 1x1 patch size for ViT: My original point may have been unclear. While the rebuttal correctly state that pixel-level patchification (i.e. 1x1 patches) is implemented at lower input resolutions, the critique was centered around the discrepancy between the title's claim ("An Image Is Worth 50,176 Tokens") and the actual evaluations. Specifically, table 1 does not present results for a sequence length of 50,176 on DeiT-Base trained on ImageNet, despite this being a central message of the paper. Based on the runtime extrapolation from Table 6, training with this sequence length would take approximately 6.6 years on a single A100 GPU (80GB) with a batch size of 5 (a figure that is already optimistic). Given the substantial computational demands, I strongly recommend including a carbon footprint section. Such a section would contextualize the environmental implications of the proposed approach and help readers understand the trade-offs involved. This could be benchmarked using standard ImageNet training protocols on A100 or similar hardware.
> >
> > 2. “The overall method… showing results on vision and timeseries tasks.” This is an error on my part. This reference was mistakenly carried over from another paper I reviewed that included timeseries experiments. I apologize for the oversight. Rest assured that this error does not influence my evaluation of the paper and is only a typographical error.
> >
> > 3. “without ever indulging into the GLOPs, test and training time compute”. While this was initially an oversight, revisiting Figure 3 reaffirms my concern. The method demonstrates only marginal performance gains despite incurring approximately 150x the computational cost in FLOPs. This trade-off raises serious concerns about the practicality of the approach. For example, training a single epoch with batch size 1 requires 967 GPU hours, compared to only 0.36 GPU hours for DeiT-Base. This stark difference highlights a major limitation of the proposed method despite increased hardware capabilities.
> >
> > I appreciate the answers provided for Q3 and Q4 in the sense that the answers are reasonable. Regarding answer to Q5, it still leaves the primary objective of the paper ambiguous. If the intention is to present an analytical perspective on patch size scaling, then the paper should frame itself explicitly as such, rather than prescribing a specific patchification strategy (e.g. pixel-level patchification or "learning from pixels") as a new norm due to increased hardware capabilities. "For fixed architecture, improving accuracy from 82.6 to 84.6 is a significant achievement." "Siginifcant" is a subjective term but it must be weighed against the massive increase in FLOPs (from 1x to 150x). These gains do not come at zero cost. To improve transparency, I suggest reporting the GFLOPs alongside the metrics in the main results table (e.g. Tab 1), allowing readers to better evaluate the trade-offs involved.
> >
> > Unanswered Question: Why is the paper using the altered ICML template (e.g. the footer is missing) and the whole **Impact statement section is missing**? The impact statement section for the proposed patchification laws is needed even more due to the gigantic increase in computational demands.
> >
> >  My main concerns are the framing of the paper as not being analytical enough, alteration of the template for unfair space gains compared to other papers, not being transparent enough about the impact of the proposed patchification "laws" with exponential compute increase, no theorectical discussion.
> > I am open to reevaluating my rating based on the answers provided to me and other reviewers. Based on the answers provided so far to me and other reviewers, i will increase my rating to 2.

---

> > > ### Author Response · Authors · 2025-04-08
> > >
> > > We sincerely appreciate your detailed comments and the improved scores. We are pleased to see that some issues have been resolved, and here we provide further explanations to address your remaining concerns:
> > >
> > > ***Accuracy-Computation Trade-off:*** For all studies investigating scaling laws, their greatest value lies in revealing the potential of scaling in a specific direction—that is, along a particular dimension, how much gain can be achieved. With the model being scaled-up, the accuracy-computation trade-off will become inevitably worse as we are approaching its performance limits. Therefore, assessing a scaling direction solely based on the growth of FLOPs is unfair. A more objective comparison can be achieved by contrasting one scaling dimension against another. For example, in Figure 3, we compare patch size scaling with traditional parameter scaling, where we not only achieve a better trade-off but also demonstrate a superior scaling limit, suggesting that we already have favorable scaling performance in terms of both efficiency and effectiveness.
> > >
> > > Furthermore, "scaling law", as a widely accepted term, refers to studies aimed at exploring the scaling-up potential of a specific dimension within a given architecture rather than proposing new structures or methods, so our work is an analytical study. For a scaling law, empirical results are the most direct (or almost the only) means of assessment, as our goal is to demonstrate its practical performance. Like most scaling laws, a model's actual capacity is difficult to prove mathematically, but empirically fitted scaling curves (as shown in our Figures 1 and 3) already provide sufficient insight to the community.
> > >
> > > ***ViT with 1x1 Patch Size:*** Thank you for your updated question. Our paper aims to explore whether patchification granularity can be a reliable scaling direction in visual understanding. Using the linear complexity of Adventurer, we have drawn a positive conclusion and demonstrated that an image can be scaled to 50,176 tokens, which we believe is an exciting finding and thus included it in the title. For ViTs, the scaling conclusions obtained on smaller inputs are consistent with those of Adventurer. While ViT, due to its higher complexity, demands more computational power for processing very long sequences, this is an intrinsic limitation of ViT itself, not a weakness of our study. Conversely, our evaluations help the community better understand this limitation: to handle longer sequences, ViT needs to introduce essential lower-level optimizations such as FlashAttention, SplashAttention, and KV-sharing strategies. With these strategies, the actual runtime of ViT is significantly shorter, and we have roughly estimated that an experiment with ViT at 224x224 resolution and 1x1 patch size could be completed within a week using 512 TPU v4 cores.
> > >
> > > ***ICML Footer:*** In the official ICML template, whether to display the footer depends on the command \printAffiliationsAndNotice{}, which is commented out by default in the template, so the footer does not display. We have inquired with the officials, and they indicated that this is permitted during the review phase.
> > >
> > > ***Computation Cost and Environmental Impact:*** Thank you for your suggestion; we will specify the specific resource consumption such as FLOPs and runtime in more parts of the paper (e.g., Table 1). We apologize for overlooking the impact statement section, as we thought it was optional. We will include the following statement in the final version:
> > >
> > > *This paper presents work whose goal is to advance the field of Machine Learning. Our experiments involved approximately 50,000 A100 GPU hours, which is considered a modest level of resource consumption compared to large-scale vision or language model research. While there are many potential societal consequences of our work, none are significant enough to warrant specific highlighting in this context. We believe the ethical impacts and societal implications are well-aligned with the advancement of machine learning technology.*

---

### Official Review · Reviewer_DwN7 · 2025-03-18

**Overall Recommendation:** 4

**Summary:**

The paper addressed the scalability issue of patch sizes in vision transformer (ViT), which is a widely-adopted backbone in vision-related task. In past research a moderate parameter of patch size is used by default when ViT is chosen as the backbone. This study empirically investigate the effect of varying patch size (e.g., reducing it down to even 1*1) and find a scaling-law like rule in a variety of computer vision tasks (recognition, detection, segmentation).

## update after rebuttal

Thanks for the response and additional experiments to clarify my concern on SwinTransformer. I have also checked the comments from other reviewers and stick to my original recommendation.

**Claims And Evidence:**

All are fine except for possibly problematic setting in the section "limitations of input size scaling" and Figure 4. The comparisons may not be on some fair basis.

**Essential References Not Discussed:**

n/a

**Ethical Review Concerns:**

There is no ethical issue found in the submission.

**Experimental Designs Or Analyses:**

In fact most of pages in the paper were devoted to experiments, including both the reported performance scores on the chosen benchmarks and a series of ablation studies to reveal the effect of key factors (e.g., scaling of patch or parameter). I carefully check the experimental settings. They seem following previous practice, with sufficient details presented.

**Methods And Evaluation Criteria:**

The evaluations are conducted on several most popular benchmarks (e.g., ImageNet, COCO etc.) and with standard metrics (e.g., average precision for object detection and instance segmentation). I have not concern about the evaluations.

**Other Comments Or Suggestions:**

n/a

**Other Strengths And Weaknesses:**

Overall I regard this is a good work with clear insight and experimental designs. The key idea (varying patch size to investigate whether and kind of scaling law holds) seem empirically validated by a series of experimental results in this work. As far as I know such a study is still missing in the literature, thus I currently lean to recommend to accept this work.

However, there is still some space for improving the work. One key difference between computer vision and other general tasks lies in the sparsity of attentions in ViT. One example is the swinTransformer as proposed by Microsoft research. The work reveals the scalability of increasing visual token-based sequences. It is not clearly discussed whether the claims still hold when it is combined with sparse and spatially local attentions. I would suggest the authors to include additional experiments and discussion.

The section "limitations of input size scaling" is not reasonable. There are some key technical details missing in the main text, particularly the specific way of increasing the number of parameters in patchification. Fixing the ratio of image-size / patch-size makes patches from different images not granularity-equal. According to my experiences, this will complicated the training of the model since it has to tackle more complex input during generalization. All above make the claim not fully convincing.

**Questions For Authors:**

Please read me comments on the weakness of this work.

**Relation To Broader Scientific Literature:**

The work reveals the importance of choosing proper patch size in using ViT. Given the popularity of ViT in a large number of domains (computer vision, medical image analysis, weather forecasting, earthquake prediction etc.), the insight reported here may be valuable in improving many models nowadays used in these domains.

**Theoretical Claims:**

There is no theoretic proof or claim. This is a work fully based on empirical evaluations on data.

---

> ### Author Rebuttal · Authors · 2025-04-01
>
> We appreciate the reviewer’s constructive feedback and detailed suggestions. The detailed response to your questions/concerns are presented below:
>
> **Q1: *Additional experiments and discussion about sparse and spatially local attention networks.***
>
> **A1:** Thank you for your constructive suggestion. We followed your feedback and conducted additional experiments on Swin Transformer, with the summarized results provided below. Specifically, we use Swin-T as the base model, which has an initial patch size of 4×4 and hierarchically downsamples the feature map by a final factor of 8 in both width and height during intermediate stages. We scaled it up by reducing the initial patch size to 2×2 and 1×1, while keeping the default downsampling rates unchanged. This scaling strategy closely mirrors the approach we used when handling standard ViT models. As shown in the table, our conclusion also holds for Swin Transformer—smaller patch sizes consistently lead to lower test loss and improved accuracy. In the revision, we will include larger local-attention models such as Swin-S and Swin-B to further support our findings.
>
> | Initial patch size | Inference time | Test loss | Accuracy |
> |--------------------|----------:|----------:|---------:|
> | 4x4                |       1x  |     0.806 |    81.3  |
> | 2x2                |     3.9x  |     0.734 |    82.0  |
> | 1x1                |    15.1x  |     0.697 |    82.5  |
>
> **Q2: *Limitations of input size scaling.***
>
> **A2**: We appreciate your insightful comments on this matter. We would like to explain the motivation behind this set of experiments: in fact, input size scaling and patch size scaling are largely equivalent—they both proportionally change the feature processing granularity and the sequence length of the model. Through these experiments, our goal is simply to show that patch size is a more suitable choice for a standardized direction of scaling, since it does not affect input storage, and input size scaling does not necessarily offer performance advantages over patch size scaling beyond a certain range.
>
> Specifically, we know that increasing the input size within a certain range tends to improve model performance. For example, on ImageNet, using the same model such as DeiT-Base/Patch16, an input resolution of 384×384 achieves noticeably better accuracy compared to 224×224 (e.g., 83.1 vs. 81.8). We believe this improvement basically comes from two sources: 1)  the direct benefit of increased computation from scaling up the input; and 2) a higher input resolution reduces the distortion caused by resizing images during preprocessing. To ablate these two effects, we keep a fixed ratio between input size and patch size, which helps maintain a roughly constant computational cost and figure out the impact of the second term. Then our experiment demonstrates that this benefit tends to vanish once the input resolution exceeds the average original image size in the dataset.
>
> We will carefully rephrase the description and analysis of this experiment in the revised version based on your suggestions and comments.

---

### Decision · Program_Chairs · 2025-05-01

**Decision:**

Accept (poster)

**Comment:**

The paper received mixed reviews, with the two of reviewers (DwN7, UjKW) recommending acceptance, and two reviewers (vQ14, KY13) suggesting rejection. The AC has thoroughly examined the submitted materials, including the paper itself, the reviewers’ concerns and the authors’ responses, and has carefully considered all factors in the decision-making process.

To the AC, the main concern is on the limited utility of the patch size scaling law, as the paper shows (in Fig. 4) further scaling the input size beyond original image size is no longer helpful; therefore compared to model size scaling, there is a potential hard limit in this scaling axis, which compromises the promise to further scale. Nevertheless, the work has done a solid exploration along this direction, already adopting an efficient architecture to mitigate the computational concern, and the overall value is deemed outweighing the concerns which are mostly addressed by the responses.

Therefore recommending acceptance. Please incorporate promised changes in the final version.